# SageAttention: Accurate 8-bit attention for Plug-and-Play Inference Acceleration

**Jintao Zhang, Jia Wei, Pengle Zhang, Jun Zhu, Jianfei Chen**\*
Dept. of Comp. Sci. & Tech., Institute for AI, BNRist Center,
Tsinghua-Bosch Joint ML Center, THBI Lab, Tsinghua University
{zhang-jt24@mails., jianfeic@, dcszj@}tsinghua.edu.cn

## Abstract

The transformer architecture predominates across various models. As the heart of the transformer, attention has a computational complexity of $O(N^2)$, compared to $O(N)$ for linear transformations. When handling large sequence lengths, attention becomes the primary time-consuming component. Although quantization has proven to be an effective method for accelerating model inference, existing quantization methods primarily focus on optimizing the linear layer. In response, we first analyze the feasibility of quantization in attention detailedly. Following that, we propose SageAttention, a highly efficient and accurate quantization method for attention. The OPS (operations per second) of our approach outperforms FlashAttention2 and xformers by about **2.1x** and **2.7x**, respectively. SageAttention also achieves superior accuracy performance over FlashAttention3. Comprehensive experiments confirm that our approach incurs almost **no end-to-end metrics loss across diverse models**—including those for large language processing, image generation, and video generation. The code is available at https://github.com/thu-ml/SageAttention.

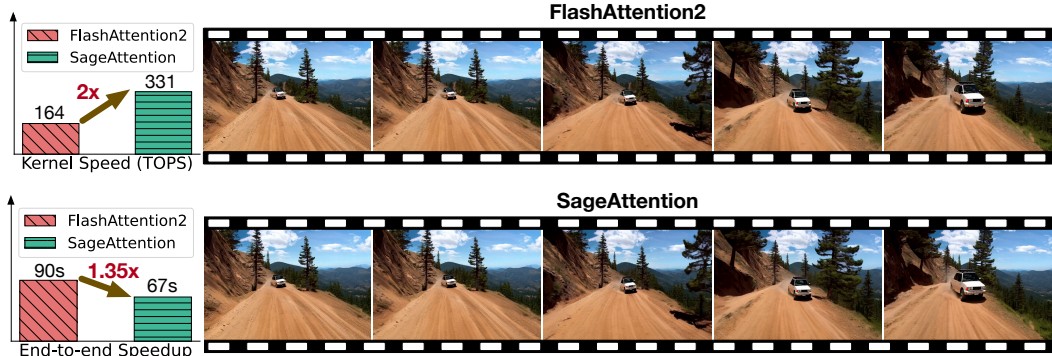

Figure 1: An example of SageAttention on video generation (CogvideoX on RTX4090).

## 1 Introduction

Attention is the fundamental component of transformers (Vaswani, 2017), and efficiently computing attention is crucial for transformer-based applications. Moreover, there is a recent trend in processing longer sequences, which further strengthens the need for faster attention. In tasks like video generation (Yang et al., 2024) and language model prefilling (Dubey et al., 2024), the sequence length can easily go up to 8K∼ 128K. Due to its quadratic complexity, the cost of attention dominates all other operations in such scenarios, as illustrated in Figure 2.

Quantization is an effective strategy for enhancing neural networks' computational and memory efficiency by reducing the numerical precision. There are abundant works on accelerating training (Sun et al., 2019; Xi et al., 2024b; Peng et al., 2023) and inference (Jacob et al., 2018; Xiao et al., 2023a)

---
\*Corresponding author.

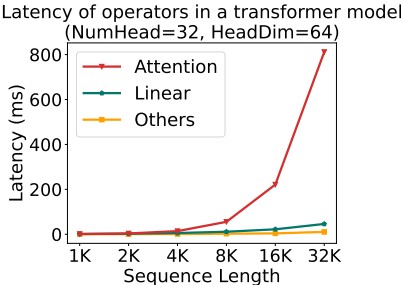

Figure 2: Latency of attention.

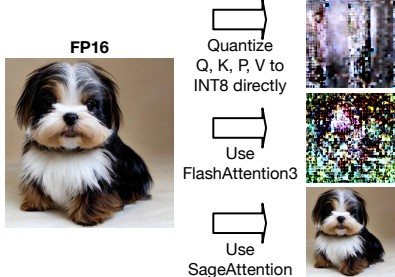

Figure 3: A comparison example.

with low-precision numerical formats such as FP8, INT8, or INT4. However, existing works primarily focused on quantizing the *linear* layer, where *attention* is left unaccelerated in high-precision, such as FP16. There is not yet a work that systematically investigates the quantization of attention. Moreover, many quantization methods require extra training, and the cost can be prohibitive for large-scale models. While FlashAttention3 (Shah et al., 2024) was released recently and offers an FP8 version, it is tailored to and can only be used with the Nvidia Hopper architecture. This exclusive optimization limits its broader applicability. Furthermore, our analysis demonstrates that directly implementing the FP8 version can lead to performance degradation, as detailed in Table 1.

Quantizing attention is challenging. The computation of attention is more complex than that of linear operations. Attention includes a softmax operation and two matrix multiplication (Matmul) operations: $QK^\top$ and $PV$. Direct 8-bit quantization and dequantization of the matrices $(Q, K, P, V)$ in attention will result in significantly degraded performance across various models. For example, the text-to-image model Unidiffuser (Bao et al., 2023) will generate a completely blurry image with both INT8 and FlashAttention3's FP8 implementation (See Figure 3), and Llama2 only achieves a random-guessing-level accuracy of 25.5% on the MMLU dataset with INT8 attention. After investigating deeply, we identified two primary challenges: **(C1)** The matrix $K$ exhibits a significant channel-wise outlier, leading to substantial accuracy loss during quantization. **(C2)** Simply quantizing $(P, V)$ into INT8 does not consistently ensure the accuracy of $PV$ across various scenarios.

In this paper, we propose `SageAttention`, a quantization method to accelerate attention while *preserving accuracy*. `SageAttention` is easy-to-use. As a post-training quantization method, it can be *used in a plug-and-play manner in inference time* by simply replacing the original high-precision implementation. We propose several techniques to achieve this goal. First, we opt to quantize the tensors in attention to INT8 rather than FP8. This decision is based on the fact that INT8 Matmul on some commonly used GPUs, e.g., RTX4090 and 3090, are four times faster than in FP16 and two times faster than FP8. Moreover, INT8 quantization for matrices $(Q, K)$ is more precise than FP8 in attention (See Table 2). To address **(C1)**, we propose a method to smooth the $K$ matrix. This method significantly enhances accuracy with a negligible time overhead (<0.2%). To address **(C2)**, as an alternative to quantizing $(P, V)$ to 8-bit, we propose a more accurate yet efficient method for the Matmul $PV$: we maintain $(P, V)$ in FP16 and use a low-precision FP16 accumulator. This strategy doubles Matmul's speed without sacrificing any accuracy. Finally, we implement several versions of attention with different speed-accuracy tradeoffs and propose a method to select the fastest attention implementation for each layer while preserving accuracy.

We offer a high-performance implementation of `SageAttention` on RTX4090 and 3090 GPUs using Triton (Tillet et al., 2019). Our implementation contains a fused kernel combining ROPE with quantization and a fast self-attention kernel inspired by FlashAttention-style tiling. The implementation utilizes the fast INT8 *mma(u8.u8.s32)* and FP16-with-FP16-accumulator *mma(f16.f16.f16)* instructions of Nvidia Tensor Core. Our kernel is about 2.1× and 2.7× faster than FlashAttention2 and xformers, respectively. Notably, it achieves 340 TOPS on RTX4090 at headdim=64 and headdim=128, reaching 52% of the theoretical INT8 throughput. In contrast, the peak for the state-of-the-art FlashAttention2 is only 165 TOPS. Moreover, at headdim=64, our throughput on RTX 4090 is even close to the 490 TOPS throughput of FlashAttention3, which is exclusive to the much more powerful and expensive Hopper GPUs. We extensively evaluate the end-to-end metrics of our approach on state-of-the-art image/video generation, image classification, and language models. On all tasks, `SageAttention` can be directly adopted in a plug-and-play manner with negligible loss in model performance, while offering more than 2× speedup than FlashAttention2 and xformers.

## 2 RELATED WORK

We categorize efficient Attention works into three groups: **(1) Sparse Attention.** This strategy only selects parts of a sequence from a given context for processing with standard Attention. Implementations like Swin transformer (Liu et al., 2021), Twins (Chu et al., 2021), UniFormer (Li et al.), Attentionsinks (Xiao et al., 2023b), InfLLM (Xiao et al., 2024), LongLora (Chen et al., 2023), Minference (Jiang et al., 2024), and SkipAttention (Venkataramanan et al., 2023) show promise. However, these methods' limitations are that they only work in a few scenarios because omitted calculations are not always useless. **(2) Linear Attention.** Techniques that transform Attention computation to reduce time complexity, for example, Linformer (Wang et al., 2020), Performer (Choromanski et al., 2020), MetaFormer (Yu et al., 2022), and LinearAttention (Katharopoulos et al., 2020), which lower the time complexity of Attention from $O(N^2)$ into $O(N)$. These methods excel in specific scenarios while standard Attention remains prevalent. **(3) Kernel Optimization.** Rather than simplifying calculations, these methods exploit hardware capacities to enhance speed. The xformers (Lefaudeux et al., 2022) platform accelerates Attention with customizable blocks and dedicated CUDA kernels. FlashAttention (Dao et al., 2022) proposes tiling to reduce the memory reads/writes between GPU global memory and on-chip SRAM for significant speedups. FlashAttention2 (Dao, 2023) refine the parallelism and warps partition of FlashAttention. Bikshandi & Shah (2023) further optimize FlashAttention2 by kernel fusion. FlashAttention3 (Shah et al., 2024) is proposed for Hopper architecture. However, FlashAttention3 is exclusive to the Hopper GPU architecture, and the accuracy of its quantization version is significantly lower than our method (See Table 1). RingAttention (Liu et al.) scales FlashAttention across multiple GPUs. I-bert (Kim et al., 2021) quantizes all tensors in a transformer block into INT8 but is restricted to RoBERTa. Our method falls under the third category, and is **orthotopic** with the first and second categories.

## 3 PRELIMINARY

Our method builds on FlashAttention-2 and adopts dynamic quantization. We will begin by reviewing FlashAttention-2, followed by a brief introduction to dynamic quantization techniques.

### 3.1 FLASHATTENTION

The computation of self-attention can be formulated as follows: $S = QK^\top/\sqrt{d}$, $P = \sigma(S)$, $O = PV$, where $\sigma(S)_{ij} = \exp(S_{ij})/\sum_k \exp(S_{ik})$ is the softmax operation. The matrices $Q$, $K$, and $V$ each have dimensions $N \times d$, while the matrices $S$, $P$ are $N \times N$. While $d$ is typically small, e.g., 64 or 128, $N$ can be thousands if not millions. Therefore, the $N \times N$ matrices $(S, P)$ are much larger than $(Q, K, V)$, and a naive implementation suffers from the huge amount of global memory I/O for $(S, P)$ reads/writes. FlashAttention (Dao, 2023) proposes to tile $Q$, $K$, and $V$ from the token dimension into blocks $\{Q_i\}, \{K_i\}, \{V_i\}$ with block sizes of $b_q, b_{kv}, b_{kv}$, respectively. Then, to avoid the memory I/O for $(S, P)$, it uses online softmax (Milakov & Gimelshein, 2018) to progressively compute each block of $O$, i.e., $O_i$ as follows.

First, for each block of $\{K_i\}, \{V_i\}$, it computes the following equations iteratively:

$$S_i^j = Q_i K_j^\top/\sqrt{d}, \quad (m_i^j, \widetilde{P}_i^j) = \tilde{\sigma}(m_i^{j-1}, S_i^j), \tag{1}$$

$$l_i^j = \exp(m_i^{j-1} - m_i^j)l_i^{j-1} + \mathrm{rowsum}(\widetilde{P}_i^j), \quad O_i^j = \mathrm{diag}\left(\exp(m_i^{j-1} - m_i^j)\right)O_i^{j-1} + \widetilde{P}_i^j V_j \tag{2}$$

Where $m_i^j$ and $l_i^j$ are $b_q \times 1$ vectors, which are initialized to $-\infty$ and 0 respectively. $\tilde{\sigma}()$ is an online softmax operator: $m_i^j = \max\{m_i^{j-1}, \mathrm{rowmax}(S_i^j)\}$, $\widetilde{P}_j^i = \exp(S_i^j - m_i^j)$.

Finally, after all iterations, i.e., $j = b_{kv}$, the output $O_i$ can be computed by $O_i = \mathrm{diag}(l_i^j)^{-1}O_i^j$.

### 3.2 DYNAMIC QUANTIZATION

A matrix multiplication $C = AB$ can be accelerated with quantization as:

$$(\delta_A, \hat{A}) = \psi(A), \quad (\delta_B, \hat{B}) = \psi(B), \quad \hat{C} = \hat{A}\hat{B}, \quad C = \psi^{-1}_{\delta_A \delta_B}(\hat{C}) \tag{3}$$

Here, $\psi$ is a *quantizer* which converts a high-precision (e.g., FP32) matrix $A$ to a low-precision format $\hat{A}$ (e.g., INT8 or FP8) with a *scale* $\delta_A$, and $\psi^{-1}$ is a *dequantizer* to convert back to high-precision. We should have $\psi_{\delta_A}^{-1}(\hat{A}) \approx A$. The actual matrix multiplication $\hat{A}\hat{B}$ is carried in low-precision. In modern GPUs, low-precision matrix multiplication is usually multiple times faster than higher-precision ones.

Many quantizers depend on the numerical format and granularity, e.g., how many elements share a common scale factor. For example, an INT8 *per-tensor dynamic quantizer* first computes the scale as the maximum absolute value of the entire tensor, scales the elements to the maximum representable range of INT8 [-127, +127], and then casts to INT8 with rounding: $\hat{A} = \lceil A/\delta_A \rfloor, \delta_A = \max(|A|)/127$. Likewise, *per-token quantizer* assigns a scale factor for each token of a tensor: $\hat{A}[i,:] = \lceil A[i,:]/\delta_A \rfloor, \delta_A[i,:] = \max(|A[i,:]|)/127$. Also, *per-channel quantizer* assigns a scale factor for each channel of the tensor, i.e., along the channel dimension: $A[:,i] = \lceil A[:,i]/\delta_A \rfloor, \delta_A = \max(|A[:,i]|)/127$. Based on the tiling approach of FlashAttention, we can apply per-block quantization correspondingly. *per-block quantizer* assigns a scale factor for every $b = m - n$ tokens: $\hat{A}[m:n,:] = \lceil A[m:n,:]/\delta_A \rfloor, \delta_A = \max(|A[m:n,:]|)/127$. Dequantization simply involves a element-wise scaling: $\psi_{\delta_A}^{-1}(\hat{A}) = \delta_A \hat{A}$.

# 4 SAGE ATTENTION

In this section, we propose `SageAttention`, a fast yet accurate method to accelerate attention computation with 8-bit quantization. Considering that most networks are not natively trained with quantized attention, `SageAttention` is designed to be plug-and-play.

Unlike linear layers, which are easy to quantize, quantizing attention is more complicated. Extra treatment is required to ensure both good accuracy and fast speed. First, we will formulate quantized attention in Section 4.1, followed by introducing our approach.

## 4.1 FORMULATION

Based on the description of FlashAttention and dynamic quantization in Section 3.1 and 3.2, we formulate the quantized attention as follows.

**Quantization:** $(\delta_Q, \hat{Q}) = \psi_Q(Q/\sqrt{d}), \ (\delta_K, \hat{K}) = \phi_K(K), \ (\delta_P, \hat{P}) = \psi_P(\widetilde{P}), \ (\delta_V, \hat{V}) = \psi_V(V)$ (4)

**Attention:** $S = \psi_{\delta_Q \delta_K}^{-1}(\hat{Q}\hat{K}^\top), \ (m', P) = \tilde{\sigma}(m, S), \ O = \mathrm{diag}\left(\exp(m' - m)\right)O + \psi_{\delta_P \delta_V}^{-1}(\hat{P}\hat{V})$ (5)

$\phi_K$ is a transformation to obtain quantized $K$, which we shall discuss in subsequent sections. For simplicity, we omit all superscripts and subscripts, but the matrices used in attention are still tiles, and the computation is still organized as FlashAttention described in Section 3.1. Compared to the original full-precision version, as shown in Eq. 4, 5, `SageAttention` adds quantizers to $Q, K, P, V$ and dequantizers to the product to accelerate both Matmuls of $QK^\top$ and $PV$. Online softmax is left in full-precision.

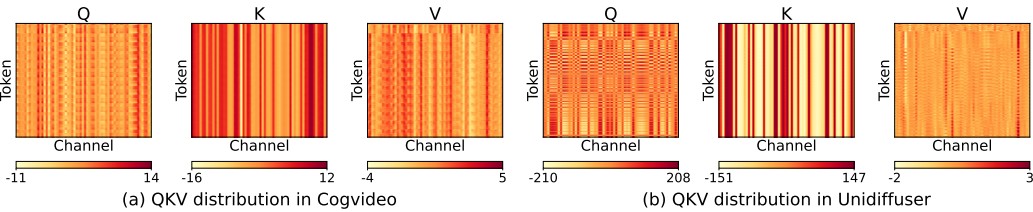

Figure 4: Typical examples of data distribution of (Q, K, V).

Table 1: End-to-end metrics comparison of different quantization methods.

| Quantization (Q, K) | Smoothing K | Llama WikiText ↓ | CogVideo (Fscore) ↑ | Unidiffuser (FID) ↓ | UltraPixel (FID) ↓ | TIMM ImageNet ↑ |
|---|---|---|---|---|---|---|
| **Full-Precision** | - | 5.823 | 3.768 | 163.33 | 179.78 | 84.79% |
| **Per-token** | ✗ | 5.824 | 1.924 | 221.18 | 193.36 | 84.21% |
| | ✓ | 5.824 | **3.734** | **166.52** | **179.79** | 84.74% |
| **Per-block** | ✗ | 5.825 | 2.014 | 229.08 | 195.67 | 84.18% |
| | ✓ | 5.824 | **3.718** | **166.93** | **179.98** | 84.76% |
| **Per-tensor** | ✗ | 5.826 | 1.902 | 267.06 | 196.26 | 84.12% |
| | ✓ | 5.824 | **3.640** | **167.65** | **180.21** | 84.69% |
| **FlashAttn3 (with quant)** | | 5.850 | 3.394 | 394.13 | 383.61 | 84.70% |

## 4.2 SMOOTH MATRIX K

Directly quantizing $Q, K$ often results in a large error. Particularly, quantizing $Q, K$ to INT8 yields completely blurry image/video in text-to-image/video tasks. As shown in Figure 4.1, we visualize two typical groups of $Q, K, V$ from a text-to-image model `Unidiffuser` (Bao et al., 2023) and a text-to-video model `CogvideoX` (Yang et al., 2024). Notably, $K$ exhibits distinct channel-wised outliers. However, per-channel quantization cannot be applied for $K$, because quantization can only be performed at the outer axis (token dim) of the Matmul $QK^\top$. Moreover, the previous smoothing technique proposed for linear layers (Xiao et al., 2023a) cannot be applied since $Q$ is also heavily affected by outliers. Fortunately, the channel outliers of $K$ have a pattern: Each token's key is actually a *large bias shared by all tokens*, plus a small token-wise signal. Therefore, the outlier is not from large variation across tokens, but simply the large bias. Based on this observation, we propose to smooth the matrix $K$ by a transform $\gamma$, which subtracts averaged $K$ across all tokens:

$$\gamma(K) = K - \text{mean}(K) \tag{6}$$

where $\text{mean}(K) = \frac{1}{N} \sum_{t=1}^{N} K[t,:]$ is the average key, with a shape $1 \times d$. Note that such a transformation does not change the attention score $P$, because for any query $q$, we have $\sigma(q(K - \text{mean}(K)^\top)) = \sigma(qK^\top - q \cdot \text{mean}(K)) = \sigma(qK^\top)$. Finally, the transformation from full-precision $K$ to quantized $\hat{K}$ can be written as $\phi_K(K) = \psi_K \circ \gamma$, where $\psi_K$ is a quantizer. In other words, a full-precision $K$ is substracted with the mean, before eventually being quantized.

Table 1 presents end-to-end metrics for different quantization methods with and without *smoothing K* on various models. The results demonstrate that *smoothing K* offers significant benefits of accuracy. Moreover, the speed overhead of smoothing $K$ for attention is less than **0.2%** (See Table10).

Table 2: **Average accuracy** using different data types across all layers of real models.

| $Q, K$ | $\widetilde{P}, V$ | Cos Sim ↑ | Relative L1 ↓ | RMSE ↓ |
|---|---|---|---|---|
| **INT8** | E4M3 | 99.94% | 0.0345 | 3.53e-3 |
| | E5M2 | 99.81% | 0.0572 | 6.11e-3 |
| | INT8 | 99.70% | 0.1035 | 6.82e-3 |
| E4M3 | E4M3 | 99.81% | 0.0607 | 5.93e-3 |
| | E5M2 | 99.68% | 0.0769 | 7.72e-3 |
| | INT8 | 99.58% | 0.1199 | 8.31e-3 |
| E5M2 | E4M3 | 99.37% | 0.1107 | 1.09e-2 |
| | E5M2 | 99.22% | 0.1213 | 1.20e-2 |
| | INT8 | 99.13% | 0.1583 | 1.24e-2 |

Table 3: **Worst accuracy** using different data types across all layers of real models.

| $Q, K$ | $\widetilde{P}, V$ | Cos Sim ↑ | Relative L1 ↓ | RMSE ↓ |
|---|---|---|---|---|
| **INT8** | E4M3 | 76.36% | 0.5899 | 0.4311 |
| | E5M2 | 78.98% | 0.4233 | 0.4371 |
| | INT8 | 56.40% | 0.7921 | 0.5405 |
| | **FP16** | 99.99% | 0.0116 | 0.0091 |

## 4.3 QUANTIZATION FOR Q, K, P, V

**Quantization granularity for** $Q, K$: $\psi_Q(Q)$ and $\psi_K(K)$ can be set with the granularity of per-token, per-block or per-tensor. This is because per-channel quantization is not feasible, since the scale factors of the inner axis of $QK^\top$ cannot be used to do dequantization (Xiao et al., 2023a).

**Data type of** $Q, K$: We choose INT8 for $\psi_Q(Q)$ and $\psi_K(K)$ for two reasons. First, Table 2 shows the average accuracy using different data types (INT8, E4M3, E5M2) for $Q, K, \widetilde{P}, V$ across all layers of `Llama2` (7B) (Touvron et al., 2023) and `Unidiffuser`. It shows that quantizing $Q, K$ to INT8 performs higher accuracy than using E4M3 and E5M2. Second, Matmul using INT8 is two times faster than using FP8 in many commonly used GPUs, e.g., RTX4090 and 3090.

**Quantization granularity for** $\widetilde{P}, V$: We propose to use $\psi_P(\widetilde{P})$ in per-block and $\psi_V(V)$ in per-channel for three reasons. (1) Per-channel quantization for $\widetilde{P}$ and per-token quantization for $V$ are not viable because dequantization requires scale factors of the outer axis. (2) $\widetilde{P} = \exp(S_i - \text{rowmax}(S_i))$, where $S_i$ is the Matmul result of a block of $Q$ and $K^T$, the max value in each row of $\widetilde{P}$ is 1. Hence, we can assign a single static scale $s = \frac{1}{127}$ to a block $\widetilde{P}$, whose accuracy equals per-token quantization. (3) Per-channel quantization can address the channel-wised outlier of $V$.

**Data type of** $\widetilde{P}, V$: We choose INT8 for $\psi_P(\widetilde{P})$ and $\psi_V(V)$ because Matmul using INT8 is two times faster than using FP8 in some commonly used GPUs, and although the accuracy using $\psi_P(\widetilde{P})$ and $\psi_V(V)$ in INT8 is worse than E4M3 and E5M2, the average accuracy is similar (See Table 2).

**Accuracy metrics.** We use three metrics to assess the accuracy of quantized attention output $O'$ compared to attention output in full-precision $O$: First, we flatten $O'$ and $O$ into vectors in the shape of $1 \times n$. Then, Cosine Sim$= \sum OO' / \sqrt{\sum O^2} \sqrt{\sum O'^2}$, Relative L1$= \sum |O - O'| / \sum |O|$, RMSE$= \sqrt{(1/n) \sum (O - O')^2}$.

Table 4: **Average accuracy** using different accumulators across all layers of real models.

| Accum. | Cos Sim ↑ | Relative L1 ↓ | RMSE ↓ |
|---|---|---|---|
| **FP32** | 99.98% | 0.0156 | 2.94e-3 |
| **FP16** | 99.98% | 0.0156 | 2.94e-3 |

Table 5: **Worst accuracy** using different accumulators across all layers of real models.

| Accum. | Cos Sim ↑ | Relative L1 ↓ | RMSE ↓ |
|---|---|---|---|
| **FP32** | 99.84% | 0.0511 | 4.229e-3 |
| **FP16** | 99.84% | 0.0511 | 4.229e-3 |

### 4.4 FP16 ACCUMULATOR: MUCH MORE ACCURATE AND EFFICIENT SOLUTION

The above solution for $\psi_P(\widetilde{P})$ and $\psi_V(V)$ has one problem, that is, the accuracy using INT8 is very poor in some model layers. Table 3 shows the worst accuracy using different data types for $Q, K, \widetilde{P}, V$ across all layers of `Llama2` and `Unidiffuser`. It shows that INT8 $\psi_P(\widetilde{P})$ and $\psi_V(V)$ bring an unacceptable error. In response, we propose a very accurate and also efficient solution. Specifically, we propose to use FP16 as the data type of Matmul $\widetilde{P}V$ with an FP16 accumulator.

The benefit of such a solution is obvious. First, in the context of some commonly used GPUs, e.g., RTX4090 and 3090, the speed of Matmul in FP16 with an FP16 accumulator is **2x** faster than that with an FP32 accumulator. Moreover, using FP16 accumulators can save more register resources than using FP32 accumulators, accelerating the computation speed. Second, Table 3 shows that using FP16 for $\widetilde{P}, V$ is much more accurate than using all the other 8-bit data types. Moreover, using FP16 accumulators incurs no accuracy loss than using FP32 accumulators. Specifically, Table 4 and 5 show the average and worst accuracy using FP16 or FP32 accumulators on all layers of `Llama2` and `Unidiffuser`, showing that there is no accuracy loss of using the FP16 accumulator.

Table 6: Four kernel implementations of `SageAttention`.

| Kernel | $\psi_Q(Q), \psi_K(K)$ | $\psi_P(P)$ | $\psi_V(V)$ |
|---|---|---|---|
| `SAGEAttn-T` | per-token, INT8 | FP16, FP16 Accumulator | FP16, FP16 Accumulator |
| `SAGEAttn-B` (Algorithm 1) | per-block, INT8 | FP16, FP16 Accumulator | FP16, FP16 Accumulator |
| `SAGEAttn-vT` (Figure 5(a)) | per-token, INT8 | per-block, INT8 | per-channel, INT8 |
| `SAGEAttn-vB` | per-block, INT8 | per-block, INT8 | per-channel, INT8 |

### 4.5 ADAPTIVE QUANTIZATION

Based on the discussion in Section 4.3 and 4.4, we implement four attention kernels (See Table 6) based on two sets of choices: (1) Using $\psi_Q(Q)$ and $\psi_K(K)$ in per-token or per-block.

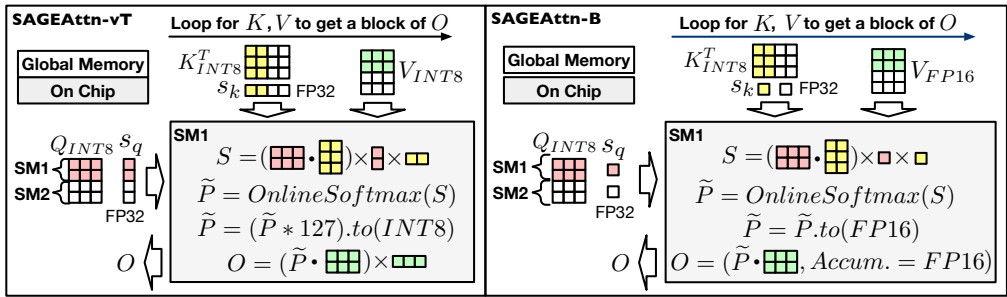

(a) SageAttention (per-token quantize Q,K; INT8 V)     (b) SageAttention (per-block quantize Q,K; FP16 V)

Figure 5: Workflow of SageAttention.

---

**Algorithm 1:** Implementation of `SAGEAttn-B`.

---

**Input:** Matrices $Q(FP16), K(FP16), V(FP16) \in \mathbb{R}^{N \times d}$, block size $b_q, b_{kv}$.

**Preprocessing:** $K = K - \text{mean}(K)$; // Subtracting the mean value across tokens

**Quantization:** $(\delta_Q, \hat{Q}) = \psi_Q(Q/\sqrt{d}), \ (\delta_K, \hat{K}) = \psi_K(K)$; // INT8 per-block quant

Divide $\hat{Q}$ into $T_m = N/b_q$ blocks $\{\hat{Q}_i\}$, and divide $\hat{K}, V$ into $T_n = N/b_{kv}$ blocks $\{\hat{K}_i\}$ and $\{V_i\}$;

**for** *i in [1, $T_m$]* **do** ;    // Outer loop is paralleled in SMs (stream processors)

    Load $\hat{Q}_i$ and $\delta_Q[i]$ into a SM ;

    **for** *j in [1, $T_n$]* **do**

        Load $\hat{K}_j, V_j$, and $\delta_K[j]$ into the SM ;

        $S_i^j = \text{Matmul}(\hat{Q}_i, \hat{K}_j^T) \times \delta_Q[i] \times \delta_K[j]$;

        $m_i^j = \max(m_i^{j-1}, \text{rowmax}(S_i^j)), \widetilde{P}_i^j = \exp(S_i^j - m_i^j), l_i^j = e^{m_i^{j-1} - m_i^j} + \text{rowsum}(\widetilde{P}_i^j)$ ;

        $O_i^j = \text{diag}(e^{m_i^{j-1} - m_i^j})^{-1} O_i^{j-1} + \text{Matmul}(\widetilde{P}_i^j.\text{to(FP16)}, V_j, \text{Accum\_type = FP16})$ ;

    $O_i = \text{diag}(l_i^{T_n}) O_i^{T_n}$ ;

    Write $O_i$ ;

**return** $O = \{O_i\}$;

---

(2) Using $\psi_P(\widetilde{P})$ and $\psi_V(V)$ in INT8 or retaining $\widetilde{P}, V$ in FP16 with an FP16 accumulator. `SAGEAttn-B` is accurate enough for all models and can achieve a 2x speedup (See Figure 6 and 7). However, `SAGEAttn-vB` is also accurate for some layers in a model and faster a little (about 4%) than `SAGEAttn-B`. Therefore, we use various inputs to test the cosine similarity of `SAGEAttn-vB` for each layer of a model. Then, we will select `SAGEAttn-vB` for those layers where `SAGEAttn-vB`'s cosine similarity is bigger than 99.8% (the worst similarity of `SAGEAttn-B`), and the other layers are left for `SAGEAttn-B`.

### 4.6 FUSION TRICKS AND PERFORMANCE ANALYSIS

**Fusion Tricks.** To reduce the overhead of quantization, we fuse the quantization process with the operator preceding the attention layer. For instance, we fuse quantization within the ROPE (Rotary Position Embedding) (Su et al., 2021) layer. Specifically, before the ROPE result ($A$) is written from shared memory into global memory, we perform $\delta_A, \hat{A} = \psi(A)$. Subsequently, the $\delta_A, \hat{A}$ are written into global memory. Additionally, we also fuse the coefficient $(1/\sqrt{d})$ of $QK^T$ into the quantization process rather than leaving it in the attention layer. Specifically, we multiply $Q$ by $(1/\sqrt{d})$ on chip before quantizating $Q$.

**Performance Analysis.** We will take `SAGEAttn-B` as an example to discuss the acceleration effects on actual hardware: (1) Matmul acceleration. Utilizing INT8 matrix multiplication units on current mainstream hardware can achieve **2-4×** throughput. While FP16 accumulators do not offer throughput improvements on most compute cards, on-edge accelerators, such as the RTX4090, can still achieve a 2x improvement over FP32 accumulators. (2) Quantization overhead. Quantization and dequantization are considered the main overhead in current quantization methods (Lin et al., 2024). The computational overhead can not be avoided, but through fusing the quantization of $Q, K$ with ROPE, we avoid the IO overhead of quantization. (3) Cache and registers. Currently,

mainstream accelerators need to store data in a cache (such as SharedMemory) during computation. Using 8-bit data for calculations can reduce the usage of the general cache, and using fp16 accumulators can also reduce the usage of accumulation registers. (4) Dram access. Using 8-bit data can halve the tensors transfer overhead from DRAM to the compute units. Although quantization introduces additional FP32 scales, these scales can be considered negligible compared to the tensors.

## 5 EXPERIMENTS

**Main results.** The speed of `SageAttention` is approximately **2.1×** faster than FlashAttention-2. Furthermore, `SageAttention` achieves an average real speedup of **2.83×** compared to the original attention in various models, with **negligible loss in end-to-end metrics**.

### 5.1 EXPERIMENTAL SETUP

**Models.** We validate the effectiveness of `SageAttention` across a diverse set of representative models from the fields of language, image, and video generation. Specifically, we conduct experiments on five models: `Llama2` (7B) (Touvron et al., 2023) for text2text, `CogvideoX` (Yang et al., 2024) for text2video, `Unidiffuser` (Bao et al., 2023) and `UltraPixel` (Ren et al., 2024) for text2image, `TIMM` (Wightman, 2019) for image classification, and `Llava1.6` (Liu et al., 2024a) for visual question answering.

**Datasets.** `Llama2` is evaluated on three zero-shot tasks: WikiText (Merity et al., 2022) to assess the model's prediction confidence, LAMBADA (Paperno et al., 2016) evaluate contextual understanding, and MMLU (Hendrycks et al., 2020) for measuring knowledge across various subjects. `CogvideoX` is evaluated using the open-sora (Zheng et al., 2024c) prompt sets. Both `UltraPixel` and `Unidiffuser` are assessed on the COCO annotations (Lin et al., 2014), featuring (prompt, image) pairs. `TIMM` is evaluated on on three image datasets: ImageNet (Deng et al., 2009), ImageNet-Sketch (Sketch) (Wang et al., 2019), and ImageNet-Rendition (ImageNet-r) (Hendrycks et al., 2021). `Llava1.6` is evaluated on three datasets: TextVQA (Singh et al., 2019), POPE (Li et al., 2023b), and VQAv2 (Goyal et al., 2017).

**Metrics.** For `Llama2`, we use perplexity (ppl.) (Jelinek et al., 1977) for WikiText, and Accuracy (Acc.) for LAMBADA and MMLU. For `CogvideoX`, folowing (Zhao et al., 2024a), we evaluate the quality of generated videos on five metrics: CLIPSIM and CLIP-Temp (CLIP-T) (Liu et al., 2024b) to measure the text-video alignment; (VQA-a) and (VQA-t) to assess the video aesthetic and technical quality, respectively; and Flow-score (FScore) for temporal consistency (Wu et al., 2023). For `UltraPixel` and `Unidiffuser`, generated images are compared with the images in the COCO annotations dataset in three aspects: FID (Heusel et al., 2017) and sFID (Salimans et al., 2016) for fidelity evaluation, *Clipscore* (CLIP) (Hessel et al., 2021) for text-image alignment, and *ImageReward* (IR) (Xu et al., 2024) for human preference. For `TIMM` and `Llava1.6`, we use Accuracy.

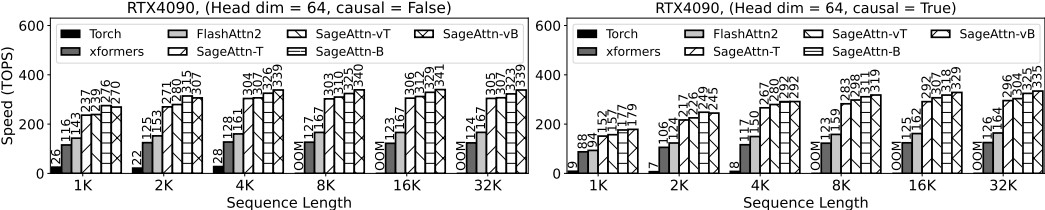

Figure 6: Speed comparison between `SageAttention` and baselines (RTX4090, headdim=64).

### 5.2 SPEED AND ACCURACY OF ATTENTION KERNELS

**Speed.** We conduct experiments to compare the Speed of `SageAttention` against baselines using configurations with headdim=64 or headdim=128, both with and without Causal Mask Vaswani (2017). Specifically, Figure 6 and Figure 7 show the Speed of `SageAttention` and baselines across varying sequence lengths on RTX4090. These results indicate that `SageAttention`

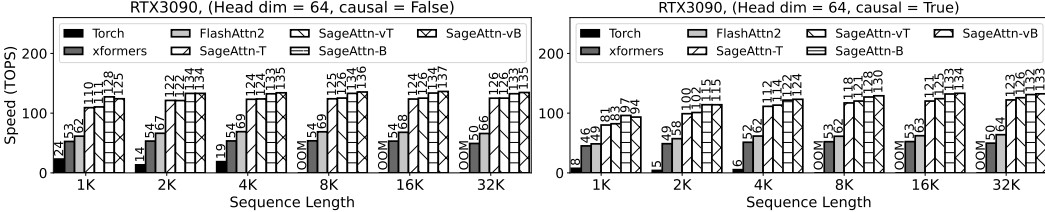

Figure 7: Speed comparison between `SageAttention` and baselines (RTX4090, headdim=128).

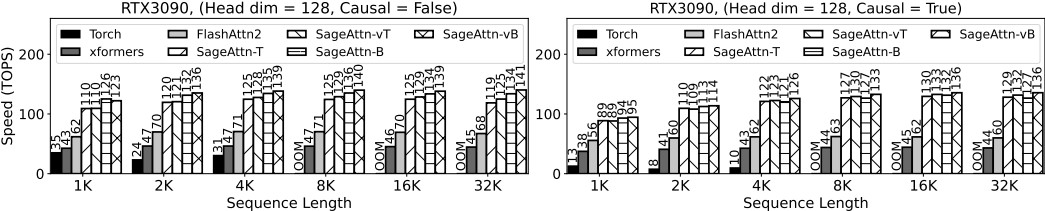

Figure 8: Speed comparison between `SageAttention` and baselines (RTX3090, headdim=64).

Figure 9: Speed comparison between `SageAttention` and baselines (RTX3090, headdim=128).

achieves a peak of **341** TOPS and is **2x** faster than FlashAttention2 and **2.9x** faster than xformers on average. Figure 8 and Figure 9 illustrate the results on RTX3090, showing a similar speedup performance.

**Accuracy.** Table 9 shows the numerical error of four implementations of `SageAttention` compared with attention in full-precision. This experiment is conducted using a set of (Q, K, V) conforming to a normal distribution. It shows the error of the four implementations is rather small. `SAGEAttn-T` and `SAGEAttn-B` achieve 100% cosine similarity and RMSE in the e-4 level.

Table 7: Real speedup of `SageAttention` on RTX4090.

| Model | Shape of Q, K, V | Original attention | SageAttention | Speedup |
|---|---|---|---|---|
| CogvideoX | (2, 30, 17776, 64) | 163.37 (FlashAttn2) | **327.57** | **2.01x** |
| Llama2 | (4, 32, 1536, 128) | 130.99 (FlashAttn2) | **231.74** | **1.77x** |
| UltraPixel | (2, 32, 7285, 64) | 152.03 (FlashAttn2) | **325.18** | **2.14x** |
| Unidiffuser | (4, 24, 1105, 64) | 105.68 (xformers) | **246.93** | **2.34x** |
| TIMM | (12, 64, 197, 64) | 18.910 (Torch) | **111.41** | **5.89x** |

## 5.3 END-TO-END PERFORMANCE

**Speedup.** We measure the real speed of `SageAttention` and the original attention on `Unidiffuser`, `UltraPixel`, `CogvideoX`, `Llama2` and `TIMM` on RTX4090. Table 7 shows that `SageAttention` outperforms original attention across all models. Specifically, `SageAttention` yields **2.83x** speedup compared to the original attentions on average.

**Metrics loss.** We assessed the end-to-end metrics of various models using `SageAttention` compared to using attention in full-precision. Detailed evaluation results are presented in Table 8 for `Llama2`, `CogvideoX`, `Unidiffuser`, `UltraPixel`, and `TIMM`, respectively. The results indicate that `SageAttention` successfully matches the performance of attention in full-precision across all models. Specifically, on `Llama2`, `CogvideoX`, `UltraPixel`, and `Unidiffuser`,

Table 8: End-to-end metrics loss across text, image, and video generation models.

| Model | Attention | WikiText (Ppl.) ↓ | Lambda (Acc.) ↑ | MMLU (Acc.) ↑ |
|---|---|---|---|---|
| Llama2 | Full-Precision | 5.823 | 0.886 | 0.46 |
| | **SageAttention** | **5.824** | **0.887** | **0.46** |

| Model | Attention | CLIPSIM ↑ | CLIP-T ↑ | VQA-a ↑ | VQA-t ↑ | FScore ↑ |
|---|---|---|---|---|---|---|
| CogvideoX | Full-Precision | 0.1837 | 0.9976 | 68.962 | 75.925 | 3.7684 |
| | **SageAttention** | **0.1836** | **0.9976** | **68.839** | **75.037** | **3.8339** |

| Model | Attention | FID ↓ | sFID ↓ | CLIP ↑ | IR ↑ |
|---|---|---|---|---|---|
| Unidiffuser | Full-Precision | 163.33 | 145.08 | 0.3152 | 0.1609 |
| | **SageAttention** | **166.49** | **143.18** | **0.3154** | **0.1521** |
| UltraPixel | Full-Precision | 179.78 | 141.35 | 0.3132 | 0.6169 |
| | **SageAttention** | **179.79** | **141.63** | **0.3131** | **0.6110** |

| Model | Attention | ImageNet (Acc.) ↑ | Sketch (Acc.) ↑ | ImageNet-r (Acc.) ↑ |
|---|---|---|---|---|
| TIMM | Full-Precision | 84.79% | 45.32% | 59.55% |
| | **SageAttention** | **84.74%** | **45.78%** | **60.32%** |

| Model | Attention | TextVQA (Acc.) ↑ | POPE (Acc.) ↑ | VQAv2 (Acc.) ↑ |
|---|---|---|---|---|
| Llava1.6 | Full-Precision | 60.25% | 86.45% | 77.55% |
| | **SageAttention** | **60.09%** | **86.44%** | **77.47%** |

SageAttention resulted in only a minor average degradation of 0.2% compared to attention in full-precision. Moreover, on TIMM, SageAttention even surpasses attention in full-precision.

Table 9: Accuracy of SageAttention kernels.

| attention | Cos Sim ↑ | Relative L1 ↓ | RMSE ↓ |
|---|---|---|---|
| SAGEAttn-T | 1.0 | 0.019 | 6.8e-4 |
| SAGEAttn-B | 1.0 | 0.021 | 7.3e-4 |
| SAGEAttn-vT | 99.9% | 0.064 | 0.065 |
| SAGEAttn-vB | 98.9% | 0.138 | 0.067 |

Table 10: Overhead of smoothing K.

| Model | Smooth K | TOPS ↑ |
|---|---|---|
| CogvideoX | ✗ | 327.57 |
| | ✓ | **327.52** |
| UltraPixel | ✗ | 325.18 |
| | ✓ | **324.56** |

Table 11: Benefit of adaptive quantization.

| attention | model | CLIPSIM ↑ | TOPS ↑ | Model | MMLU ↑ | TOPS ↑ |
|---|---|---|---|---|---|---|
| SAGEAttn-T | CogvideoX | 0.1827 | 292.17 | Llama2 | 0.46 | 208.59 |
| SageAttention | | **0.1835** | **327.57** | | **0.46** | **231.74** |

## 5.4 ABLATION STUDY

**Overhead of smoothing K.** Table 10 presents the overhead associated with smoothing K on the attention speed in real models. The results indicate a minimal reduction, less than 0.2%.

**Benefit of adaptive quantization.** We analyzed the performance differences between using only SAGEAttn-T and employing an adaptive strategy (SageAttention). Table 11 presents the metrics and average speed of attention on CogvideoX and Llama2. The results indicate that the adaptive strategy increases the speed of attention by 11.7% without any loss in metrics.

## 6 CONCLUSION AND FUTURE WORK

We introduce SageAttention, an efficient and precise INT8 quantization method for attention. First, we propose a method to smooth matrix K, enhancing the accuracy with under 0.2% speed overhead. Second, we use FP16 accumulators in the Matmul of (P, V) to boost both accuracy and speed. Third, we use adaptive quantization to further improve OPS by 12% without sacrificing accuracy. Our method surpasses FlashAttention2 and xformers by approximately **2.1x** and **2.7x**, respectively. Extensive testing confirms that our approach maintains end-to-end metrics across various models, including language, image, and video generation models.

ACKNKOWLEGEMENT

The authors would like to thank Haofeng Huang for his valuable help with the implementation. This work was supported by the NSFC Project (No. 62376131), Tsinghua Institute for Guo Qiang, and the High Performance Computing Center, Tsinghua University. J.Z is also supported by the XPlorer Prize.

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

# A  EXPERIMENTAL DETAIL

## A.1  ENVIRONMENT

We implemented our Attention kernels using OpenAI Triton (Tillet et al., 2019) and conducted experiments on Ubuntu 22.04 servers. Tests on the RTX 4090 utilized a server with PCIE 5.0, a 16-core Xeon(R) 6430 CPU, and 120GB DDR4 RAM, while the RTX3090 tests employed a server with a 16-core Xeon(R) 8358P CPU and 80GB DDR4 RAM. To reproduce our results, experiments should be conducted in the environment of torch 2.4.0+cu121, triton-nightly (version of 20240816), python 3.11, and (gcc, g++) in version 9.

## A.2  HYPER-PARAMETERS FOR ATTENTION KERNELS

We use 128 for a block size of $Q$, and 64 for a block size of $K$ and $V$. The parameters Num_Warps and Num_Stages, which represent the number of warp schedulers and the number of processing stages in our GPU kernels, respectively, are detailed in Table 12.

Table 12: Hyper-parameters for our Attention Kernels.

| HeadDim | Causal Mask | Num_Warps | Num_Stages |
|---------|-------------|-----------|------------|
| 64 | False | 4 | 3 |
| 64 | True | 4 | 4 |
| 128 | False | 8 | 3 |
| 128 | True | 8 | 5 |

## A.3  DETAILS OF DATASETS AND MODELS

We choose the first 256 annotations from the COCO 2014val dataset as the prompt set for `UltraPixel` and `Unidiffuser` image generation. We also used the corresponding 256 images of the 256 prompts as the ground truth images to calculate the FID and sFID. For `CogvideoX`, the model is trained on long texts, so we applied an open-sora prompt set, each consisting of more than 120 words. The specific model we used for `TIMM` is *vit_base_patch16_224.augreg2_in21k_ft_in1k*.

## A.4  ADDITIONAL EXPERIMENTS

Table 13: Comparison of SageAttention with AWQ (W4A16) on Llama2.

| | Full-Precision | SageAttention | AWQ | AWQ+SageAttention |
|---|---|---|---|---|
| **Perplexity↓** | 5.4721 | **5.4729** | 5.5988 | 5.5998 |
| **Speedup of Linear Computation** | 0 | 0 | 0 | 0 |
| **Speedup of Attention** | 0 | **2x** | 0 | **2x** |

Table 14: Comparison of SageAttention with Q-diffusion (W8A8) on Unidiffuser.

| | FID↓ | sFID↓ | CLIP↑ | ImageReward↑ |
|---|---|---|---|---|
| **Full Precision** | 163.33 | 145.08 | 31.52 | 0.1609 |
| **SageAttention** | **166.49** | **143.18** | **31.54** | **0.1521** |
| **Q-diffusion (W8A8)** | 395.99 | 178.56 | 18.03 | -2.273 |

## A.5  COMPARISON WITH OTHER METHODS

There are some task-specific quantization methods, such as AWQ for LLMs, Q-diffusion for text-to-image, and ViDiT-Q for text-to-video applications. SageAttention is orthogonal to them because those works are mainly used to quantize the linear layers. Second, AWQ is only used to compress the parameters of LLMs with no acceleration effect in computation. Q-diffusion has not reported its

Table 15: Comparison of SageAttention with VIDIT-Q on CogvideoX.

| | CLIPSIM↑ | CLIP-T↑ | VQA-a↑ | VQA-t↑ | FScore↑ | End-to-end Speedup↑ |
|---|---|---|---|---|---|---|
| **Full Precision** | 0.1837 | 0.9976 | 68.962 | 75.925 | 3.7684 | - |
| **SageAttention** | 0.1836 | 0.9976 | 68.839 | **75.037** | 3.8339 | **34.3%** |
| **VIDIT-Q (W8A8)** | 0.1884 | 0.9974 | 68.185 | 71.011 | 3.7342 | 22% (theoretical maximum) |

acceleration effect in their paper and provided codes with acceleration effect in its official repository. ViDiT-Q has not provided the codes with acceleration effect in its official repository. Nonetheless, we compare SageAttention with those works as follows. (1) We compare the perplexity of Llama2-7B on WikiText and the speedup in the prefilling stage. The results are shown in Table 13. We compare SageAttention with Q-diffusion (W8A8) on Unidiffuser. The results are shown in Table 14. We compare SageAttention with VIDIT-Q on CogvideoX and the results are shown in Table 15. Since the official repository does not provide acceleration code, we estimate a theoretical maximum: the Linear layer accounts for 24% of Cogvideo's latency, and W8A8 offers **at most** 4x speedup for the Linear layer, resulting in a theoretical maximum end-to-end speedup of $\frac{100}{100-24\times\frac{3}{4}}\% = 22\%$.

### A.6 SOME INSIGHTS

Table 1 shows that the metric of Llama2 remains stable with quantization. The reason is that the distribution of $Q, K$, and $V$ in the attention of Llama2-7B is relatively uniform. As a result, quantizing $Q, K$, and $V$ to INT8 or FP8 does not significantly impact the accuracy of attention. This insight inspires the idea that better control over outlier activations in models could lead to more precise quantization results. As works like Fu et al. (2024a); Zhang et al. (2024; 2025a), we believe SageAttention can also be effectively applied to various applications related to Transformers, such as MOE systems (Wang et al., 2025), linear layer quantization (Hu et al., 2025; Zhang et al., 2025e; Zhao et al., 2024a), RAG systems (Zhang et al., 2025b), training optimization (Li et al., 2023a; 2025; Huang et al., 2024; Xi et al., 2024a), heterogeneous GPU systems (Jiang et al., 2025a; Jiang et al.; 2025b), and diffusion models (Zheng et al., 2024a;b; 2025; Fu et al., 2024b; Zhao et al., 2024b; Xi et al., 2025; Zhang et al., 2025c;d).

Table 16: SageAttention based on Torch Attention.

| Sequence Length | Torch Attention | SageAttention based on Torch Attention |
|---|---|---|
| **1024** | 46 | 48 |
| **2048** | 42 | **55** |
| **4096** | 55 | **87** |
| **8192** | OOM | OOM |

## B IMPLEMENTATION BASED ON TORCH ATTENTION

FlashAttention is the state-of-the-art and most commonly used standard attention; another commonly used attention is Torch attention (PyTorch Contributors). We report the implementation speeds based on Torch in Table 16.

Table 17: Numerical error of $Q \cdot K$ using different type of quantization.

| Data Type | Cosine Sim | Relative L1 |
|---|---|---|
| **INT8** | **99.54%** | **0.084** |
| E4M3 | 92.83% | 0.342 |
| E5M2 | 77.95% | 0.681 |

### B.1 ADDITIONAL PRECISION COMPARISION

Table 17 shows the precision of $Q \cdot K$ using per-token quantization in different data types compared to $Q \cdot K$ in full precision. This experiment is conducted using $Q, K$ from the 24th layer of

Table 18: Error of quantized attention with or without smoothed K.

| Quantization Type | Smoothed K | Cosine Sim ↑ | Relative L1 ↓ | RMSE ↓ |
|---|---|---|---|---|
| Per-token (SAGEAttn-T) | Without | 62.24% | 1.187 | 0.294 |
| | **With** | **99.47%** | **0.045** | **0.031** |
| Per-block (SAGEAttn-B) | Without | 30.60% | 1.286 | 0.464 |
| | **With** | **99.31%** | **0.072** | **0.035** |
| Per-tensor | Without | 41.40% | 1.554 | 0.399 |
| | **With** | **98.06%** | **0.126** | **0.059** |
| FlashAttention-3 (quantized version) | | **26.76%** | **2.5354** | 0.5378 |

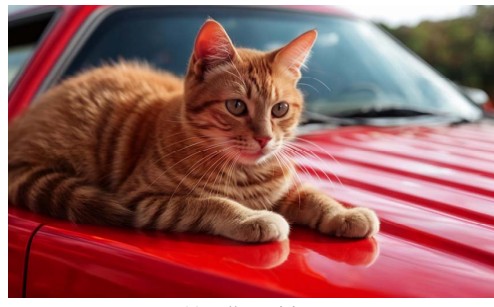

(a) Full-Precision

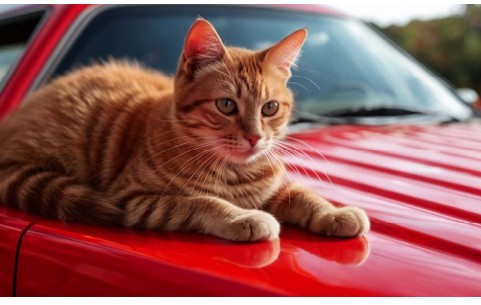

(b) SAGE Attention

Figure 10: An image generation example of UltraPixel.

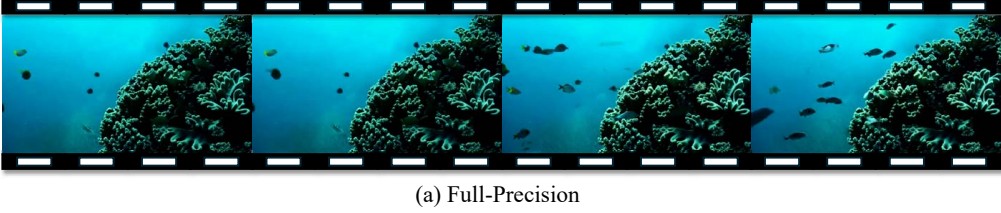

(a) Full-Precision

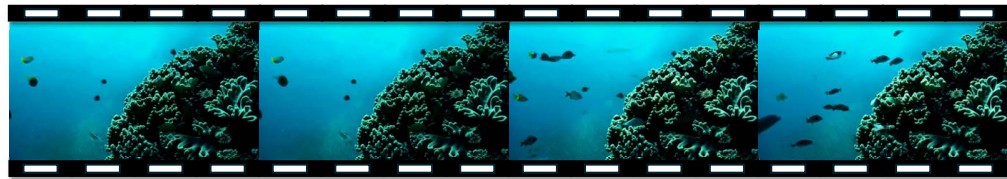

(b) SAGE Attention

Figure 11: A video generation example of Open-Sora.

Table 19: Comparison of real speedup on RTX3090.

| Model | Shape of Q, K, V | Original Attention | SAGEAttention | Speedup |
|---|---|---|---|---|
| CogvideoX | (2, 30, 17776, 64) | 71.57 (FlashAttn2) | **129.87** | **1.81x** |
| Llama2 | (4, 32, 1536, 128) | 56.54 (FlashAttn2) | **108.91** | **1.93x** |
| UltraPixel | (2, 32, 7285, 64) | 65.86 (FlashAttn2) | **131.74** | **2.00x** |
| Unidiffuser | (4, 24, 1105, 64) | 47.64 (xformers) | **108.91** | **2.29x** |
| TIMM | (12, 64, 197, 64) | 12.33 (Torch) | **66.34** | **5.38x** |

Unidiffuser. It shows that quantizing $Q, K$ to INT8 performs higher precision than using E4M3 and E5M2.

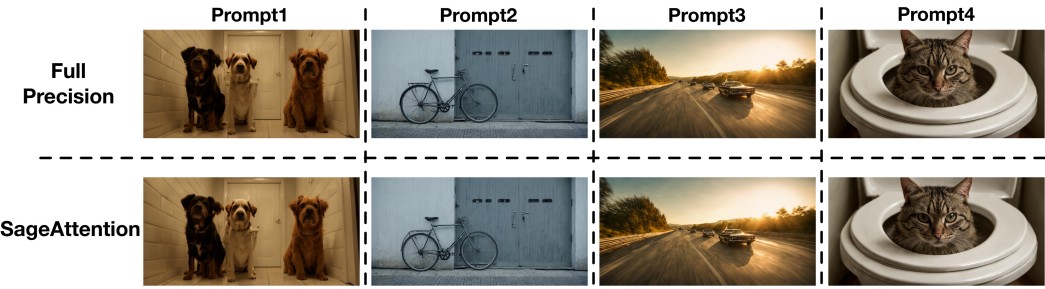

Figure 12: More image generation examples of UltraPixel, where prompt1="Two dogs are looking up while they stand near the toilet in the bathroom", prompt2="A gray bicycle is locked to some metal doors", prompt3="An image of a car driving on the highway", and prompt4="A cat on the lid of a toilet looking perturbed".

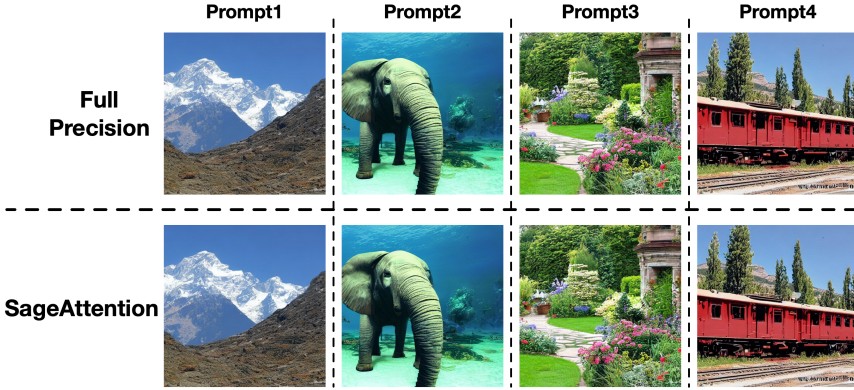

Figure 13: More image generation examples of Unidiffuser, where prompt1="Beautiful view of the Himalayas", prompt2="An elephant under the sea", prompt3="English Country Garden Design", and prompt4="An old red electric rail train in Durango, Colorado".

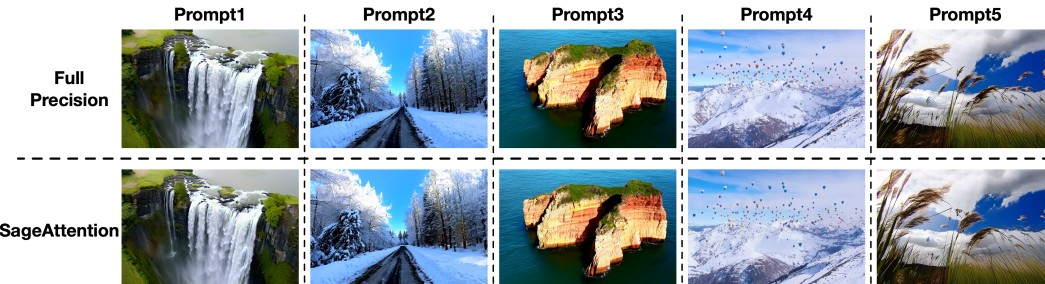

Figure 14: More image generation examples of CogvideoX. For more details about the prompts and the full videos, refer to `https://anonymous.4open.science/r/image_video_examples-3E44/README.md`.

Table 18 shows the precision of different quantization methods with and without *smoothing K* on various models. The results demonstrate that *smoothing K* offers significant benefits of precision.

### B.2 VISUALIZED RESULTS

Figure 10 shows the high-resolution images (2560x1536) generated by `UltraPixel` using Attention of full precision and `SageAttention`. It can be seen that `SageAttention` matches the full precision in the high quality and highly detailed images. Figure 11 shows the videos (720x1280) generated by Open-Sora (Zheng et al., 2024c) in different precisions. `SageAttention` yields identically the same video as the full precision one.

Figure 12, Figure 13, and Figure 14 show more visualized comparison results on `UltraPixel`, `Unidiffuser`, and `CogvideoX`.

## B.3 REAL SPEEDUP ON RTX3090

We further measure the real speed of `SageAttention` and the original Attention on `Unidiffuser`, `UltraPixel`, `CogvideoX`, `Llama2` and `TIMM` on RTX3090. Table 7 shows that `SageAttention` outperforms original attention across all models. Specifically, `SageAttention` yields $2.7\times$ speedup compared to the original Attentions on average.

