# OpenReview forum: "SageAttention: Accurate 8-Bit Attention for Plug-and-play Inference Acceleration"
_ICLR.cc/2025/Conference — ICLR 2025 Poster_

### Official Review · Reviewer_xY6z · 2024-11-03

**Soundness:** 3
**Presentation:** 2
**Contribution:** 3
**Rating:** 8
**Confidence:** 4

**Summary:**

This work explores post-training quantization of the attention mechanism. It claims that it's not well explored in the literature and the existing works focus either on quantized training or post-training quantization of non-attention layers (of course, among other attention acceleration methods, like linear attention, algorithmic accelerations, etc.). The work observed that the Keys matrix has similar values per-token, and proposed to smooth it by applying a per-token bias. It explores optimal quantization strategies and combines it with flash attention algorithmic techniques to obtain the best performance. SageAttention achieves very high reconstruction quality and does not seem to lose much performance on downstream metrics.

**Strengths:**

- The performance boosts seem to be impressive compared to
- The work carries a valuable observation that the keys matrix has huge correlation per-token. It can be valuable even outside of
- I find the evaluation section to be very thorough, including many domains (text2image, text2video, LLMs). I especially appreciate text2video (where the sequences are the longest) and performance exploration on second-tier GPUs (e.g. 3090), which are more suitable for production use cases.
- I am not an expert in quantization, but I checked some prior works and didn't find similar approaches. This makes me consider the paper as novel.
- The paper provides the source code: I have not tried running it but it is valuable.

**Weaknesses:**

- I think that writing should be improved. A table I would like to see the most is the end-to-end model speed improvement and metrics degradation after integrating the proposed attention mechanism. First, for many models accelerating the attention op itself does not necessarily lead to much improved overall speed when the sequence length is short and it's MLPs who carry the main burden. Second, it's not entirely clear when looking at various tables which variant is being used at the end (and why there are so many of them throughout the paper instead of being just in some restricted ablation section). Finally, there are quite many typos and grammatical errors, e.g.:
    - L084: "matrice" => "matrices"
    - L138 (and elsewhere): "IO" => "I/O"
    - L144: "it compute" => "it computes"
    - L142=144: missing line or equation
    - L151: "The σ()" => "The" should be ommited or it should be "The function σ()".
    - L184: "First, We" => "First, we"
    - L264-L266: ~3 typos in 3 lines
    - etc.
- There are no qualitatives provided for image/video generation.

**Questions:**

- Is there a final attention variant among the proposed ones which performs the best uniformly across all tasks?
- L154: "The expression diag(l_i^j)^{-1}" would produce inf-s since l_i^j is initialized as 0. Or how is O_i^j initialized?
- L151: What is "online softmax" (i dont see it to be defined explicitly)? How does it differ from the standard softmax?

---

> ### Author Response · Authors · 2024-11-21
> **Response to Reviewer xY6z**
>
> Dear Reviewer xY6z,
>
> Thank you for recognizing the potential and effectiveness of our work and providing constructive comments. Below, we address each point raised.
>
> ---
> **W1-(1): A table I would like to see the most is the end-to-end model speed improvement and metrics degradation after integrating the proposed attention mechanism. First, for many models accelerating the attention op itself does not necessarily lead to much improved overall speed when the sequence length is short and it's MLPs who carry the main burden.**
>
> **Reply:** Thank you for your suggestion and insightful conclusion. Table8 in our paper already displays the detailed degradation of end-to-end metrics after integrating SageAttention. We add a Table to illustrate the end-to-end speedup of various models as follows.
>
> |                                            | Llama2 | Cogvideo  | Unidiffuser | Ultrapixel | TIMM   | Llava     |
> | ------------------------------------------ | ------ | --------- | ----------- | ---------- | ------ | --------- |
> | **Original End-to-end Latency**            | 0.229s | 90s       | 2.68s       | 108s       | 0.126s | 0.395s    |
> | **End-to-end Latency with SageAttention**  | 0.225s | **67s**   | 2.54s       | 105s       | 0.121s | 0.390s    |
> | **Overall End-to-end Metrics Degradation** | **0%** | **0.24%** | **0.6%**    | **0%**     | **0%** | **0.12%** |
> | **Speed up of Attention**                  | 1.77x  | 2.01x     | 2.34x       | 2.14x      | 5.89x  | 1.87x     |
>
> ---
> We highlight two examples in the following Table that demonstrate the end-to-end latency improvements achieved by SageAttention when the sequence length exceeds 10,000.
>
> |                                            | Llama3.1 (100K Sequence Length)                              | Cogvideo (18K Seq_Len) |
> | ------------------------------------------ | ------------------------------------------------------------ | ---------------------- |
> | **Original End-to-end Latency**            | 38s                                                          | 90s                    |
> | **End-to-end Latency with SageAttention**  | **26s**                                                      | **67s**                |
> | **Overall End-to-end Metrics Degradation** | **0%** (Test on [NeedleInAHaystack](https://github.com/gkamradt/LLMTest_NeedleInAHaystack) with 100K seq_len) | **0.24%**              |
>
> ---
> **W1-(2): Second, it's not entirely clear when looking at various tables which variant is being used at the end (and why there are so many of them throughout the paper instead of being just in some restricted ablation section).**
>
> **Reply:** We apologize for any confusion caused. In all tables presented in our paper, 'SageAttention' denotes the combination of SageAttn-B and SageAttn-vB, as described in Section 4.5. Notably, SageAttn-B is already twice as fast as FlashAttention and delivers the highest accuracy. We have modified the clarification in Section 4.5. For practical applications, we recommend SageAttn-B due to its superior accuracy and speed.
>
> ---
> **W1-(3): Finally, there are quite many typos and grammatical errors.**
>
> **Reply:** Thank you for thoroughly reviewing our work and pointing out the typos. We have addressed and corrected the typos you identified, along with others we discovered. All corrections have been highlighted in purple.
>
>
> ---
> **W2: There are no qualitatives provided for image/video generation.**
>
> **Reply:** Thank you for your suggestion. We add some image and video comparison examples in the Appendix in our paper. To better show the video, we also add these examples at https://anonymous.4open.science/r/image_video_examples-3E44/README.md.
>
>
> ---
> **Q1: Is there a final attention variant among the proposed ones which performs the best uniformly across all tasks?**
>
> **Reply:** Yes, SageAttn-B consistently performs the best across all tasks. Although SageAttention combines both SageAttn-B and SageAttn-vB, it is important to note that SageAttn-vB is only 4% faster than SageAttn-B, yet it exhibits inferior accuracy. Therefore, SageAttn-B remains the preferred choice due to its superior performance.
>
> | Attention Kernel | Peak TOPS | Feasibility                     |
> | ---------------- | --------- | ------------------------------- |
> | **SageAttn-B**   | 329       | All models and tasks            |
> | **SageAttn-vB**  | 341       | Only some layers in some models |
>
> ---
> **Q2: L154: "The expression diag(l_i^j)^{-1}" would produce inf-s since l_i^j is initialized as 0. Or how is O_i^j initialized?**
>
> **Reply:** Actually, the $diag(l_i^j)^{-1}$ in L154 refers to the state after completing all iterations from Lines L144-149, i.e., when $j = b_{kv}$. Initially, only $diag(l_i^0)=0$, while all subsequent $diag(l_i^j)\neq 0$. We also clarify this point by revising this part in our paper.
>
> ---

---

> ### Author Response · Authors · 2024-11-21
> **Response to Reviewer xY6z (cont.)**
>
> **Q3: L151: What is "online softmax" (I dont see it to be defined explicitly)? How does it differ from the standard softmax?**
>
> **Reply:** We explain online softmax, and its use in FlashAttention as follows.
>
> Online softmax, introduced by Maxim Milakov and Natalia Gimelshein from NVIDIA, is a kernel fusion method designed to accelerate the softmax computation on actual hardware. For a vector $ a = \\{a_i\\},{i=1..n} $, the traditional softmax first computes $ a_{max} = \max(a) $ and then $ a' = a - a_{max} $ (to maintain numerical stability), followed by computing $\text{exp}(a') / \sum_{i=1}^n \exp(a'_i) $, which requires multiple CUDA kernel launches.
>
> ---
> Online softmax fuses the computation of $ a' = a - a_{max} $ and $\sum_{i=1}^n \text{exp}(a'_i)$ into a single CUDA kernel:
>
> The calculation process initializes with $ m_0 = -\infty $ and $ d_0 = 0 $, and iteratively updates with $ m_i = \max(m_{i-1}, a_i) $ and $ d_i = d_{i-1} \times \exp(m_{i-1} - m_i) + \exp(a_i - m_i) $. In the end, $ \text{max}(a) = m_n $ and $\sum_{i=1}^n \exp(a'_i) = d_n$, and the result can be computed by $\text{exp}(a-m_n)/d_n$. This method considers each newly read $ a_i $​​, updating the maximum and rescaling the partial sum based on the updated maximum to ensure equivalent computation.
>
> ---
> Flash Attention integrates online softmax by combining it with $QK^\top$ and $PV$ multiplications, enabling a single kernel to perform the entire attention computation. Likewise,  the $S_i^j$ in L151 corresponds to the $a_i$ in the about explanation. The rows of $S_i^j$ represent multiple vectors requiring softmax computation, which, in flash attention, corresponds to the number of tokens in a tile of $Q$. The columns of $S_i^j$ indicate the number of elements in a continuous subsequence of vector $a$ processed in parallel, corresponding to the number of tokens in a tile of $K$.
>
> ---

---

> ### Author Response · Authors · 2024-11-25
> **Sincerely looking forward to further discussions**
>
> Dear Reviewer xY6z,
>
> Thank reviewer xY6z for the valuable comments and the strong acceptance of our work. We hope our reply has addressed the concerns and brought new insights to the reviewer.
>
> In addition, please let us know if you have any other questions or concerns. Thank you very much!

---

> ### Comment · Reviewer_xY6z · 2024-11-25
>
> I am thankful to the authors for such a detailed response. I have one more remaining question: why CogVideoX is referred as "Full Precision" in Table 8? As far as their README tells, fp16 was used for 2B and bf16 for 5B. Does CogVideoX use flash attention or vanilla torch attention?
>
> Also, for the specified metrics in the first table of the above response, what metrics exactly are being used to determine metrics degradation (In Table 8, there are many metrics)? Is it average across all the metrics?

---

> ### Author Response · Authors · 2024-11-26
> **Response to Reviewer xY6z**
>
> **Q4: I have one more remaining question: why CogVideoX is referred as "Full Precision" in Table 8? As far as their README tells, fp16 was used for 2B and bf16 for 5B. Does CogVideoX use flash attention or vanilla torch attention?**
>
> **Reply:** Thank you so much for your question, and we apologize for any confusion. "Full Precision" in Table 8 refers to the original precision of the target model's attention before quantization. For the CogVideoX-2B model we used, this corresponds to FP16, as pointed out. We will modify Table 8 to specify this precision from "Full Precision" to "Origin (FP16)".
>
> CogVideoX-2B originally uses FlashAttention, which is noted in Table 7. We will revise this Table to specify the precision further.
>
>
> ---
> **Q5: Also, for the specified metrics in the first table of the above response, what metrics exactly are being used to determine metrics degradation (In Table 8, there are many metrics)? Is it average across all the metrics?**
>
> **Reply:** Table 8 includes many metrics because a single end-to-end quality metric is often not sufficient for language, image, and video generation model evaluation. For example, text-to-image generation requires assessing both image quality and the alignment between text and image. Each metric reflects a potential metrics degradation, and these are commonly used measures in the field, briefly explained in Section 5.1. The "overall degradation" mentioned in the previous reply is indeed the average of the metrics degradation for each metric listed in Table 8.
>
> ---
> Please let us know if you have any other questions or concerns. Thank you very much!

---

### Official Review · Reviewer_wwcu · 2024-11-03

**Soundness:** 3
**Presentation:** 3
**Contribution:** 2
**Rating:** 6
**Confidence:** 4

**Summary:**

This paper presents SageAttention, an innovative quantization method designed to optimize the computational efficiency of attention mechanisms in transformer models by employing an 8-bit integer (INT8) quantization. It employs the FlashAttention-wise quantization and matrix smoothing with FP16 for Matmul. With all these proposed components,  the method demonstrates improved computational performance and maintains accuracy compared to existing solutions like FlashAttention2 and xformers.

**Strengths:**

+ The paper is well-written, with a logical structure and organization that facilitates understanding.

+ SageAttention shows competitive performance, outperforming FlashAttention2 and xformers by approximately 2.1x and 2.7x, respectively.

+ The method exhibits almost no end-to-end metrics loss across a variety of models, including large language models (LLMs), text-to-image (T2I), and text-to-video (T2V).

+ The discovery of channel-wise consistency, as illustrated in Figure 4, is particularly noteworthy and adds depth to the research.

**Weaknesses:**

- The method relies heavily on FlashAttention, which may weaken its technical contribution and originality. What will the performance be if it does not employ the FlashAttention as the basis?

- The reported superiority over FlashAttention3 appears to be quite marginal, raising questions about the significance of the improvements.

- Another major weakness of this paper is that it does not compare SageAttention with other task-specific quantization methods, such as AWQ [1] for LLMs, Q-diffusion [2] for text-to-image, and ViDiT-Q [3] for text-to-video applications, which could provide a more comprehensive evaluation of its performance.

[1] AWQ: Activation-aware Weight Quantization for LLM Compression and Acceleration

[2] Q-Diffusion: Quantizing Diffusion Models

[3] ViDiT-Q: Efficient and Accurate Quantization of Diffusion Transformers for Image and Video Generation

**Questions:**

Could the authors elaborate on the "Llama" column of Table 1? Specifically, why does the number remain stable even with quantization? Understanding this aspect could provide valuable insights into the robustness of the proposed method.

---

> ### Author Response · Authors · 2024-11-21
> **Response to Reviewer wwcu**
>
> Dear Reviewer wwcu,
>
> Thank you for acknowledging our presentation, performance, and novelty. We address individual comments below:
>
> ---
> **W1: The method relies heavily on FlashAttention, which may weaken its technical contribution and originality.**
>
> **Reply:** We argue that the contributions of SageAttention are significant and distinct from FlashAttention for the following reasons:
>
> 1. The implementation of SageAttention is based on FlashAttention, but this is reasonable for the following reasons. Firstly, FlashAttention is a SOTA method designed to avoid reading/writing the Softmax matrix in attention. To conduct efficient standard Attention, using such a method is almost inevitable. Just like efficient software development relies on optimized algorithm libraries. Secondly, apart from the implementation, our method can also be applied based on other attention implementations, such as Torch. Lastly, some sparse attention works, such as [1] and [2], all based on FlashAttention, were published in NeurIPS 2024 and 2023.
>
>    [1] *MInference 1.0: Accelerating Pre-filling for Long-Context LLMs via Dynamic Sparse Attention, NeurIPS 2024.*
>
>    [2] *Fast Attention Over Long Sequences With Dynamic Sparse Flash Attention, NeurIPS 2023.*
>
> 2. Our contribution is orthogonal with FlashAttention. We systematically explore the challenges of quantizing $Q, K, P, V$ on the speed and accuracy of attention across various models, and implement efficient quantized attention applicable across multiple tasks and models. Given that FlashAttention is currently the most widely used in the industry, we chose to implement our approach based on FlashAttention. Our main contributions include proposing novel quantization methods (such as smoothing K), choosing suitable numerical formats (INT8 for $QK$, and FP16 with FP16 accumulator for $PV$), and choosing suitable quantization granularity to preserve accuracy. On the implementation side, our implementation is compatible with FlashAttention and various GPUs, achieving a high acceleration ratio while ensuring accuracy. Some contributions are highly nontrivial and original.
>
> **W1-(2): What will the performance be if it does not employ the FlashAttention as the basis?**
>
> FlashAttention is the state-of-the-art and most commonly used standard attention; another commonly used attention is [Torch attention](https://pytorch.org/docs/stable/generated/torch.nn.functional.scaled_dot_product_attention.html#torch.nn.functional.scaled_dot_product_attention). We report the implementation speeds based on Torch and FlashAttention as follows.
>
> | Sequence Length | FlashAttention | SageAttention based on FlashAttention | Torch | SageAttention based on Torch |
> | --------------- | -------------- | ------------------------------------- | ----- | ---------------------------- |
> | 1024            | 145            | 276                                   | 46    | 48                           |
> | 2048            | 151            | 315                                   | 42    | **55**                       |
> | 4096            | 161            | 326                                   | 55    | **87**                       |
> | 8192            | 164            | 325                                   | OOM   | OOM                          |
> | 16384           | 164            | 329                                   | OOM   | OOM                          |
> | 32768           | 164            | 323                                   | OOM   | OOM                          |
>
>
> ---
> **W2: The reported superiority over FlashAttention3 appears to be quite marginal, raising questions about the significance of the improvements.**
>
> **Reply:** We argue that our superiority is not marginal for the following reasons:
>
> 1. As shown in Table1, SageAttn-B outperforms FlashAttention3 by 1.1x and 2.5x on end-to-end metrics on image and video generation tasks.
> 2. Although the improvement on Llama2 and TIMM is marginal, the reason is that the two tasks are easy to attention quantization.
> 3. The limitation of FlashAttention3 is obvious; that is, FlashAttention3 can only be used on and is optimized exclusively for expensive Hopper architecture GPUs, i.e., H100 and H800 GPUs. It cannot be used on commonly used GPUs such as the RTX 4090 and A100.
> 4. Figure3 only provides one visible comparison example with FlashAttention3. To supplement this, we provide more visible examples at [https://anonymous.4open.science/r/sageattention_example-FB01/README.md], showing that FlashAttention3 fails in image and video generation visibly.
>
> ---

---

> ### Author Response · Authors · 2024-11-21
> **Response to Reviewer wwcu (cont.)**
>
> **W3: Another major weakness of this paper is that it does not compare SageAttention with other task-specific quantization methods, such as AWQ [1] for LLMs, Q-diffusion [2] for text-to-image, and ViDiT-Q [3] for text-to-video applications, which could provide a more comprehensive evaluation of its performance.**
>
> **Reply:** Thank you for your advice. First, we argue that SageAttention is **orthogonal** to AWQ, Q-diffusion, and ViDiT-Q because those works are mainly used to quantize the linear layers. Second, AWQ is only used to compress the parameters of LLMs with no acceleration effect in computation. Q-diffusion has not reported its acceleration effect in their paper and provided codes with acceleration effect in its official repository. ViDiT-Q has not provided the codes with acceleration effect in its official repository. Nonetheless, we compare SageAttention with those works as follows.
>
> We compare the perplexity of Llama2-7B on WikiText and the speedup in the prefilling stage as follows.
>
> |                                      | Full-Precision | SageAttention | AWQ    | AWQ+SageAttention |
> | ------------------------------------ | -------------- | ------------- | ------ | ----------------- |
> | **Perplexity**$\downarrow$           | 5.4721   | **5.4729**    | 5.5988 | 5.5998     |
> | **Speedup of Linear Computation**    | ❌       | ❌      | ❌      | ❌     |
> | **Speedup of Attention Computation** | ❌       | **2x**        | ❌      | **2x**    |
> ---
> We compare SageAttention with Q-diffusion (W8A8) on Unidiffuser using the same experimental setting in our paper as follows.
>
> |                        | FID$\downarrow$ | sFID$\downarrow$ | CLIP$\uparrow$ | ImageReward$\uparrow$ |
> | ---------------------- | --------------- | ---------------- | -------------- | --------------------- |
> | **Full Precision**     | 163.33     | 145.08   | 31.52  | 0.1609 |
> | **SageAttention**      | **166.49**   | **143.18**   | **31.54** | **0.1521**   |
> | **Q-diffusion (W8A8)** | 395.99   | 178.56   | 18.03  | -2.273   |
> ---
> We compare SageAttention with VIDIT-Q on CogvideoX using the same experimental setting in our paper as follows. Since the official repository does not provide acceleration code, we estimate a theoretical **theoretical maximum:** the Linear layer accounts for 24% of Cogvideo's latency, and W8A8 offers **at most** 4x speedup for Linear layer, resulting in a theoretical maximum end-to-end speedup of 100/ (100 - 24*3/4)% = 22%.
>
> |                    | CLIPSIM$\uparrow$ | CLIP-T$\uparrow$ | VQA-a$\uparrow$ | VQA-t$\uparrow$ | FScore$\uparrow$ | End-to-end Speedup$\uparrow$      |
> | ------------------ | ----------------- | ---------------- | --------------- | --------------- | ---------------- | --------------------------------- |
> | **Full Precision** | 0.1837            | 0.9976           | 68.962          | 75.925          | 3.7684           | -                                 |
> | **SageAttention**  | 0.1836         | 0.9976           | 68.839          | **75.037**      | 3.8339           | **34.3%**                         |
> | **VIDIT-Q (W8A8)** | 0.1884         | 0.9974           | 68.185          | 71.011          | 3.7342           | 22% **(theoretical maximum speedup)** |
> ---
> We emphasize that SageAttention is **orthogonal** to works that focus on quantizing Linear layers such as AWQ, Q-diffusion, and ViDiT-Q.
>
> ---
> **Q1: Could the authors elaborate on the "Llama" column of Table 1? Specifically, why does the number remain stable even with quantization? Understanding this aspect could provide valuable insights into the robustness of the proposed method.**
>
> **Reply:** Thank you for your very insightful question. The reason is that the distribution of $Q, K$, and $V$ in the attention of Llama2-7B is relatively uniform. As a result, quantizing $Q, K$, and $V$ to INT8 or FP8 does not significantly impact the accuracy of attention. This insight inspires the idea that better control over outlier activations in models could lead to more precise quantization results. We add more data distribution figures for $Q, K$, and $V$ of Layer0, Layer10, and Layer20 in Llama at [https://anonymous.4open.science/r/sageattention_example-FB01/resource/rebuttal_llama_acc.png]. We also report their attention accuracy when quantizing $Q, K$ to INT8 or FP8 as follows.
>
> | Quantization   | Cos Sim$\uparrow$ | Relative L1$\downarrow$ | RMSE$\downarrow$ |
> | -------------- | ----------------- | ----------------------- | ---------------- |
> | INT8 (Layer0)  | 99.94%            | 0.0365                  | 0.0052           |
> | E4M3 (Layer0)  | 99.95%            | 0.0307                  | 0.0046         |
> | INT8 (Layer10) | 99.51%            | 0.0922                  | 0.0064           |
> | E4M3 (Layer10) | 99.28%            | 0.1230                 | 0.0076        |
> | INT8 (Layer20) | 99.75%            | 0.0702                  | 0.0094         |
> | E4M3 (Layer20) | 99.75%            | 0.0833                 | 0.0093         |
>
> ---

---

> ### Comment · Reviewer_wwcu · 2024-11-22
>
> Thanks for the detailed response of authors. Most of my concerns are addressed and I would like to raise the score (I will give score 7 if there is such an option). I highly recommend the authors to revise the manuscript to include all the new experiments.

---

> ### Author Response · Authors · 2024-11-22
> **Thank you for raising the score!**
>
> Thank you so much for raising the score! We greatly appreciate your thoughtful suggestions, which significantly improve our work. We will revise the manuscript to include all the new experiments as recommended.

---

### Official Review · Reviewer_tyTT · 2024-11-04

**Soundness:** 3
**Presentation:** 3
**Contribution:** 3
**Rating:** 6
**Confidence:** 4

**Summary:**

This papers conducts in depth analysis of viability to quantize LLMs/Diffusion models into INT8 frameworks. It also proposes smoothing methods to alleviate the outlier pains in the QKV projection process, demonstrating viable tradeoffs. Comparisons to relevant work is strong.

**Strengths:**

- Paper is well written.
- Experiments are thorough.
- Problem is challenging.

**Weaknesses:**

- Full comparison to strong SOTA methods such as Flash attention 3, though slightly mentioned in the introduction and in Table 14, is not deeply explored.
- Only targeted 4090/3000 series GPUs - it would be recommended to be tested on stronger GPUs at server level that is facing the strongest limitations.
- It would be great to test across VLMs too.

**Questions:**

As above in weakness.

---

> ### Author Response · Authors · 2024-11-21
> **Response to Reviewer tyTT**
>
> Dear Reviewer tyTT,
>
> Thank you for recognizing the potential and effectiveness of our work and providing constructive comments. Below, we address each point raised.
>
> ---
> **W1:  Full comparison to strong SOTA methods such as Flashattention3, though slightly mentioned in the introduction and in Table 14, is not deeply explored.**
>
> **Reply:** Firstly, from an accuracy perspective, SageAttention outperforms FlashAttention3, especially in video and image generation.
>
> 1. We have already compared the end-to-end metrics of SageAttention and FlashAttention3 in Table1 in our paper, where the 4-th row is SageAttn-T, and the 6th row is SageAttn-B, across language generation, image generation, video generation, and image classification models. The results show that SageAttention outperforms FlashAttention3.
>
> 2. Figure3 provides only one visible comparison example with FlashAttention3. To supplement this, we offer more visible examples at https://anonymous.4open.science/r/sageattention_example-FB01/README.md, showing that FlashAttention3 fails in image and video generation visibly.
>
> 3. We supplement the comparison of kernel accuracy for SageAttention versus FlashAttention3 when (Q, K, V) follows a standard normal distribution $\mathcal{N}(0, 1)$, with 1% replaced by values from a normal distribution $\mathcal{N}(1, 10)$. The results are in the Table below.
>
>    | Attention           | Cos Sim$\uparrow$ | Relative L1$\downarrow$ | RMSE$\downarrow$ |
>    | ------------------- | ----------------- | ----------------------- | ---------------- |
>    | **FlashAttention3** | 98.93%            | 0.1393                  | 0.0636           |
>    | **SageAttention-T** | **99.92%**        | **0.0386**              | **0.0026**       |
>    | **SageAttention-B** | **99.89%**        | **0.0467**              | **0.0210**       |
>
> ---
> Secondly, in terms of speed and applicability, the main focus of our study is to accelerate attention on commonly used hardware, such as RTX4090, 3090, and A100. However, **FlashAttention3 can only run on expensive Hopper architecture GPUs**, i.e., H100 and H800 GPUs.
>
> 4. We compare the kernel speed of SageAttention and FlashAttention3 in the Table below.
>
> | Attention           | TOPS on RTX3090 | TOPS on RTX4090 | TOPS on A100 | TOPS on A6000 | TOPS on A800 | TOPS on H100 |
> | ------------------- | --------------- | --------------- | ------------ | ------------- | ------------ | ------------ |
> | **FlashAttention2** | 68              | 165             | 204          | 110           | 188          | 327          |
> | **FlashAttention3** | ❌               | ❌               | ❌            | ❌             | ❌            | 882          |
> | **SageAttention-B** | **134**         | **322**         | **279**      | **170**       | **267**      | 533          |
>
> ---
>
> Although FlashAttention3 is faster than SageAttention on H100, this is because its main contribution lies in utilizing the programming features of the Hopper architecture. In theory, our method can also be further optimized for the Hopper architecture.
>
> ---
> **W2:  Only targeted 4090/3000 series GPUs - it would be recommended to be tested on stronger GPUs at server level that is facing the strongest limitations.**
>
> **Reply:** Yes. Our method is also effective on A100, A6000, A800, and H100 GPUs. We compare the TOPS of different attention methods on those GPUs in the following Tables.
>
> **A100:**
>
> | Sequence Length | Torch | xformers | FlashAttention2 | SageAttention-B |
> | --------------- | ----- | -------- | --------------- | --------------- |
> | 1024            | 61    | 85       | 160             | 194             |
> | 2048            | 45    | 106      | 176             | 217             |
> | 4096            | 59    | 110      | 203             | 243             |
> | 8192            | 69    | 110      | 204             | **273**         |
> | 16384           | OOM   | 111      | 204             | **278**         |
> | 32768           | OOM   | 111      | 204             | **279**         |
>
> ---
>
> **A6000:**
>
> | Sequence Length | Torch | xformers | FlashAttention2 | SageAttention-B |
> | --------------- | ----- | -------- | --------------- | --------------- |
> | 1024            | 32    | 59       | 106             | **154**         |
> | 2048            | 29    | 61       | 109             | **162**         |
> | 4096            | 34    | 61       | 112             | **168**         |
> | 8192            | 32    | 62       | 113             | **170**         |
> | 16384           | OOM   | 61       | 112             | **170**         |
> | 32768           | OOM   | 59       | 110             | **170**         |
>
> ---

---

> ### Author Response · Authors · 2024-11-21
> **Response to Reviewer tyTT (cont.)**
>
> **A800:**
>
> | Sequence Length | Torch | xformers | FlashAttention2 | SageAttention-B |
> | --------------- | ----- | -------- | --------------- | --------------- |
> | 1024            | 63    | 100      | 185             | 241             |
> | 2048            | 60    | 99       | 184             | **260**         |
> | 4096            | 62    | 102      | 185             | **264**         |
> | 8192            | 62    | 103      | 186             | **266**         |
> | 16384           | OOM   | 103      | 188             | **267**         |
> | 32768           | OOM   | 103      | 188             | **267**         |
>
> ---
>
> **H100:**
>
> | Sequence Length | Torch | xformers | FlashAttention2 | SageAttention-B |
> | --------------- | ----- | -------- | --------------- | --------------- |
> | 1024            | 103   | 128      | 256             | **439**         |
> | 2048            | 74    | 139      | 280             | **479**         |
> | 4096            | 91    | 151      | 301             | **498**         |
> | 8192            | 110   | 155      | 316             | **527**         |
> | 16384           | OOM   | 156      | 322             | **532**         |
> | 32768           | OOM   | 157      | 327             | **533**         |
>
> ---
> While the superiority of SageAttention on these GPUs is not as high as it is on consumer-level GPUs like RTX 4090 and RTX 3090, we argue that the performance degradation occurs as follows:
>
> 1. Tensor Core: On RTX 4090 and 3090, SageAttention utilizes the INT8 Tensor Core, whose throughput is 4x faster than the FP16 Tensor Core. However, on A100, A6000, A800, and H100, this speedup is reduced to 2x.
> 2. Accumulator: The speedup of the Tensor Core with FP16 accumulator is more marginal than on RTX 4090 and 3090.
> 3. Instance optimization: Some GPU architectures have unique speed optimization methods, which we need to take advantage of in future work. For example, the H100 allows for asynchronous computation between Tensor Cores and CUDA Cores.
>
>
> ---
> **W3: It would be great to test across VLMs too.**
>
> **Reply:** Thank you for your suggestion. We add an experiment on a VLM (Llava1.6-Mistral-7B) as shown in the Table below, where the TextVQA, POPE, and VQAv2 are three commonly used datasets, and the metric is accuracy. We also add this experiment to our paper.
>
> | Attention          | TextVQA$\uparrow$ | POPE$\uparrow$ | VQAv2$\uparrow$ |
> | ------------------ | ----------------- | -------------- | --------------- |
> | **Full-Precision** | 60.25%            | 86.45%         | 77.55%          |
> | **SageAttention**  | **60.09%**        | **86.44%**     | **77.47%**      |
>
> ---

---

> ### Author Response · Authors · 2024-11-25
> **Sincerely looking forward to further discussions**
>
> Dear Reviewer tyTT,
>
> We want to thank reviewer tyTT for the valuable comments. If our reply has addressed the concerns and brought new insights to the reviewer, we will highly appreciate it if the reviewer considers raising the score.
>
> In addition, please let us know if you have any other questions or concerns. Thank you very much!

---

> ### Author Response · Authors · 2024-12-02
> **Sincerely looking forward to your feedback**
>
> Dear Reviewer tyTT:
>
> We really appreciate your insightful and constructive comments, which helped us improve this paper. We have made our maximum effort to address all your concerns. As the discussion phase deadline is approaching, we would like to ask if our responses have addressed your concerns. We’d be happy to provide further clarification if there are any remaining issues.
>
> If you feel that your concerns have been resolved, we would greatly appreciate it if you consider raising the score.
>
> ---
> Best wishes!
> The Authors

---

### Official Review · Reviewer_UTAx · 2024-11-04

**Soundness:** 4
**Presentation:** 4
**Contribution:** 4
**Rating:** 8
**Confidence:** 3

**Summary:**

This paper introduces a new attention quantization method for speeding up transformer inference. To accelerate attention, the authors propose using INT8 quantization instead of FP8 for faster matrix multiplication on GPUs, along with a method for smoothing the K matrix to improve accuracy. Instead of quantizing the P and V matrices, they maintain them in FP16 and use a low-precision FP16 accumulator for faster multiplication without accuracy loss.  Finally, they offer different speed-accuracy trade-offs and a layer-wise selection method for optimal performance. Extensive experiments have been done on different transformer models for text, image and video generation tasks. The results show that the proposed SageAttention speeds up the FlashAttention2 and xformers by more than 2 times without losing any performance.

**Strengths:**

1. The contributions of this work are well motivated.

2. The proposed method seems to be novel although I am not an expert in this field.

3. The experiments are quite extensive, covering two different GPUs (RTX4090 and RTX3090), representative models for language, image, and video generation, and a wide range of datasets.

4. The results are quite impressive, showing more than two times speedup without performance degradation.

**Weaknesses:**

1. Some design choices seem to be decided by the specific hardwares that are evaluated RTX4090 and 3090 (L271). Are those design choices also compatible with other GPUs like A100 and H100?

2. Table 7 shows that different model/task has different speedup. How is the speedup related to the specific transformer architecture, model size, and complexity of the task?

**Questions:**

I am not an expert in this field and do not have additional questions at this stage. I might have more questions at the discussion phase.

---

> ### Author Response · Authors · 2024-11-21
> **Response to Reviewer UTAx**
>
> Dear Reviewer UTAx,
>
> Thank you for recognizing the potential and effectiveness of our work and providing constructive comments. Below, we address each point raised.
>
> ---
> **W1: Some design choices seem to be decided by the specific hardwares that are evaluated RTX4090 and 3090 (L271). Are those design choices also compatible with other GPUs like A100 and H100?**
>
> **Reply:** Yes. Our method is also effective on A100, A6000, A800, and H100 GPUs. We compare the TOPS of different attention methods on those GPUs in the following Tables.
>
> **A100:**
> | Sequence Length | Torch | xformers | FlashAttention2 | SageAttention-B |
> | --------------- | ----- | -------- | --------------- | --------------- |
> | 1024            | 61    | 85       | 160             | 194             |
> | 2048            | 45    | 106      | 176             | 217             |
> | 4096            | 59    | 110      | 203             | 243             |
> | 8192            | 69    | 110      | 204             | 273             |
> | 16384           | OOM   | 111      | 204             | 278             |
> | 32768           | OOM   | 111      | 204             | 279             |
>
> ---
>
> **A6000:**
> | Sequence Length | Torch | xformers | FlashAttention2 | SageAttention-B |
> | --------------- | ----- | -------- | --------------- | --------------- |
> | 1024            | 32    | 59       | 106             | 154             |
> | 2048            | 29    | 61       | 109             | 162             |
> | 4096            | 34    | 61       | 112             | 168             |
> | 8192            | 32    | 62       | 113             | 170             |
> | 16384           | OOM   | 61       | 112             | 170             |
> | 32768           | OOM   | 59       | 110             | 170             |
>
> ---
>
> **A800:**
> | Sequence Length | Torch | xformers | FlashAttention2 | SageAttention-B |
> | --------------- | ----- | -------- | --------------- | --------------- |
> | 1024            | 63    | 100      | 185             | 241             |
> | 2048            | 60    | 99       | 184             | 260             |
> | 4096            | 62    | 102      | 185             | 264             |
> | 8192            | 62    | 103      | 186             | 266             |
> | 16384           | OOM   | 103      | 188             | 267             |
> | 32768           | OOM   | 103      | 188             | 267             |
>
> ---
>
> **H100:**
> | Sequence Length | Torch | xformers | FlashAttention2 | SageAttention-B |
> | --------------- | ----- | -------- | --------------- | --------------- |
> | 1024            | 103   | 128      | 256             | 439             |
> | 2048            | 74    | 139      | 280             | 479             |
> | 4096            | 91    | 151      | 301             | 498             |
> | 8192            | 110   | 155      | 316             | 527             |
> | 16384           | OOM   | 156      | 322             | 532             |
> | 32768           | OOM   | 157      | 327             | 533             |
>
> ---
>
> While the superiority of SageAttention on these GPUs is not as high as it is on consumer-level GPUs like RTX 4090 and RTX 3090, we argue that the performance degradation occurs as follows:
>
> 1. Tensor Core: On RTX 4090 and 3090, SageAttention utilizes the INT8 Tensor Core, whose throughput is 4x faster than the FP16 Tensor Core. However, on A100, A6000, A800, and H100, this speedup is reduced to 2x.
> 2. Accumulator: The speedup of the Tensor Core with FP16 accumulator is more marginal than on RTX 4090 and 3090.
> 3. Instance optimization: Some GPU architectures have unique speed optimization methods, which we need to take advantage of in future work. For example, the H100 allows for asynchronous computation between Tensor Cores and CUDA Cores.
> ---
> **W2: Table 7 shows that different model/task has different speedup. How is the speedup related to the specific transformer architecture, model size, and complexity of the task?**
>
> **Reply:** There is no specific relationship since the speedup refers solely to the acceleration of the attention part. This speedup is independent of the transformer architecture, model size, or task complexity, because it is only related to the attention operation, which is influenced by the original implementation, e.g., xformers or FlashAttention, and the shape of Q, K, and V. Different shapes have varying speeds primarily due to slight differences in hardware utilization caused by different sequence lengths.
>
> End-to-end acceleration mainly depends on the proportion of time spent on attention. For example, attention accounts for 52% of the time in the CogVideo, and a 2x speedup in attention would result in an overall speedup of 1.35x (calculated as 100/(100-26)). If the sequence length is short, attention will occupy a smaller fraction of the time, limiting the potential for end-to-end acceleration.
>
> ---

---

### Meta-Review · Area_Chair_7sNN · 2024-12-08

**Metareview:**

This paper introduces SageAttention, a quantization technique for transformer attention, leveraging INT8 quantization and matrix smoothing to achieve substantial speedups on GPUs. Extensive evaluations show 2.1x–2.7x acceleration over FlashAttention2 and xformers without sacrificing accuracy across diverse models and tasks, including language, image, and video generation. SageAttention also outperforms FlashAttention3 in terms of accuracy for certain tasks and applicability on broader hardware.

Strengths:
* Performance Gains: SageAttention shows impressive speed-ups across different model architectures and GPUs, with consistent accuracy across a wide range of tasks.
* Broad Applicability: The method has been tested on a diverse set of models and GPUs, suggesting a robust and versatile approach.
* Thorough Evaluation: Comprehensive experiments cover both consumer and server-grade GPUs, providing insights into performance under varied conditions.

Weaknesses:
* Dependence on FlashAttention: While implementation builds on FlashAttention, reviewers accepted the justification of its orthogonality and its necessity for state-of-the-art performance.
* Comparison with Similar Methods: There's a noted lack of direct comparison with task-specific quantization methods like AWQ, Q-diffusion, and ViDiT-Q, though the authors have addressed this in rebuttals by adding comparisons.
* Presentation: Some reviewers pointed out areas in the manuscript where clarity and grammatical accuracy could be improved.

Overall, SageAttention is a well-executed work with broad applicability in accelerating transformer-based models. The thorough responses, additional experiments, and demonstrated utility across diverse tasks and hardware strengthen its case for acceptance.

**Additional Comments On Reviewer Discussion:**

Key Points Raised by Reviewers and Authors' Responses:

Hardware Compatibility (raised by UTAx and tyTT):

Concern: Limited evaluation on RTX GPUs and lack of tests on server-grade GPUs like A100, H100.
Response: Authors provided detailed experiments on A100, A6000, A800, and H100, showing consistent improvements.
Comparison to State-of-the-Art (raised by tyTT and wwcu):

Concern: Incomplete comparison with FlashAttention3 and task-specific quantization methods (AWQ, Q-Diffusion, ViDiT-Q).
Response: Authors provided new results comparing SageAttention to FlashAttention3, AWQ, and other methods, demonstrating superior generalization and orthogonality.
Dependence on FlashAttention (raised by wwcu):

Concern: Reliance on FlashAttention could weaken technical novelty.
Response: Authors clarified their orthogonal contributions and provided additional implementation results using Torch attention, showing versatility.
Writing and Presentation (raised by xY6z):

Concern: Typos, unclear organization, and lack of qualitative examples for image/video generation.
Response: Authors corrected typos, reorganized tables, and added qualitative examples and clarifications in supplementary material.
Metrics and Model Variants (raised by xY6z):

Concern: Lack of clarity on metrics aggregation and best-performing SageAttention variant.
Response: Authors clarified the aggregation process and identified SageAttn-B as the best-performing variant across tasks.

---

### Decision · Program_Chairs · 2025-01-22

Accept (Poster)